# Non-Terrestrial Networks with UAVs: A Projection on Flying Ad-Hoc Networks

**Mahyar Nemati** *[ID], **Bassel Al Homssi** [ID], **Sivaram Krishnan, Jihong Park** [ID], **Seng W. Loke** [ID] and **Jinho Choi** [ID]

School of Information and Technology, Deakin University, Geelong, VIC 3220, Australia
* Correspondence: n.mahyar@deakin.edu.au

**Abstract:** Non-terrestrial networks (NTNs) have recently attracted elevated levels of interest in large-scale and ever-growing wireless communication networks through the utilization of flying objects, e.g., satellites and unmanned aerial vehicles/drones (UAVs). Interestingly, the applications of UAV-assisted networks are rapidly becoming an integral part of future communication services. This paper first overviews the key components of NTN while highlighting the significance of emerging UAV networks where for example, a group of UAVs can be used as nodes to exchange data packets and form a flying ad hoc network (FANET). In addition, both existing and emerging applications of the FANET are explored. Next, it provides key recent findings and the state-of-the-art of FANETs while examining various routing protocols based on cross-layer modeling. Moreover, a modeling perspective of FANETs is provided considering delay-tolerant networks (DTN) because of the intermittent nature of connectivity in low-density FANETs, where each node (or UAV) can perform store-carry-and-forward (SCF) operations. Indeed, we provide a case study of a UAV network as a DTN, referred to as *DTN-assisted FANET*. Furthermore, applications of machine learning (ML) in FANET are discussed. This paper ultimately foresees future research paths and problems for allowing FANET in forthcoming wireless communication networks.

**Keywords:** flying ad hoc networks (FANET); non-terrestrial networks (NTN); terrestrial networks; unmanned aerial vehicles (UAV)



## 1. Introduction

Non-terrestrial networks (NTN) are to provide wireless connectivity from flying objects above ground usually from space and the stratosphere. In a nutshell, NTNs involve non-terrestrial flying objects such as satellites and unmanned aerial vehicles/drones (UAVs). Figure 1 illustrates the concept of NTN. Rapid development of non-terrestrial networks (NTN) in the last decade has envisioned emerging concepts of "**integrated space-terrestrial network (ISTN)**" [1–3] and the "**Internet of space things (IoST)**" [4–6], which have the potential to leverage current mobile communication networks to deliver advanced communication services for the 6th generation (6G) in the future. A notable example is mega-constellation satellite networks such as Starlink [7] and OneWeb [8], spurred by the recent advances in NTNs. By turning these networks into the IoST one can connect the unconnected in rural, rugged, and ocean environments. In addition, by integrating NTNs with terrestrial communications, one can form ISTN provisioning connectivity not only on the ground but also in the air and space.

### 1.1. Why NTN?

It is difficult for current fifth-generation (5G) wireless technologies based on the ground to provide adequate coverage in extremely remote areas while also meeting the stringent quality-of-service (QoS) standards expected of terrestrial networks. Since flying nodes are able to function at significantly greater heights in comparison to ground-based surveying methods, the NTN with flying nodes can complement terrestrial networks by

providing widespread communication for mariners, farmers, and travelers in the air and on the ground [9].

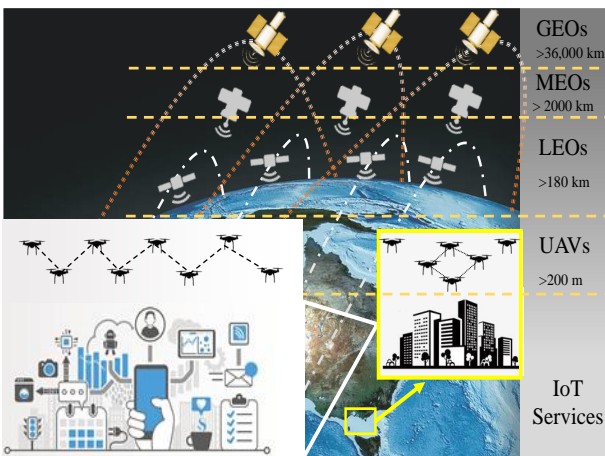

**Figure 1.** Illustration of non-terrestrial and terrestrial networks (List of abbreviations is provided at the end of the article). Although satellites are historically the first components of the NTNs, UAVs are becoming a more appealing emerging technology for integrating NTNs and terrestrial networks, e.g., in ISTN, because of their flexible flying capability at lower altitudes.

To provide fully global wireless communications everywhere and at any time, 6G wireless networks should seamlessly combine NTNs with terrestrial networks [2]. Furthermore, unlike terrestrial communications, which may experience service disruptions due to natural catastrophes or adversaries, NTNs guarantee uptime for really essential, mission-critical applications [10]. Overall, according to predictions, NTNs will boost 6G's functionality in terms of coverage, user bandwidth, system capacity, service reliability, availability, energy consumption, and connection density.

### 1.2. The Path from Satellites to UAVs

Among different types of NTN components, in this study, the UAVs are paid more attention rather than satellites. In the following, we justify why UAVs such as drones are going to become the dominant technology in NTNs and play a crucial role in many future 5G/6G and the Internet of things (IoT) applications although satellites are historically the first components of the NTNs.

There are some reasons why the market decided to take out the UAV technologies instead of only relying on satellites for future communication applications and achieving the full potential of ISTNs. For instance, the mobility of satellites in space is unidirectional and uncontrollable. Meanwhile, both terrestrial infrastructure and mobile users on the ground are relatively considered to be fixed given the very high altitudes of satellites. Given this time-varying topology, ground-to-space channels with either free space optical (FSO) or millimeter-wave (mmWave) signals are often line-of-sight (LoS) sensitive [11], so immediate transmissions may incur significant path loss. As opposed to terrestrial communications, opportunistic transmissions are not an effective solution due to the long satellite orbiting period. Handing over to other neighboring satellites is also costly due to the wireless nature of inter-satellite data backhaul links under limited energy and onboard computing resources [12].

To cope with such unfavorable topology and channel characteristics, UAVs can play a crucial role as controllable relays. In contrast to satellites and ground users, UAVs can move in any direction at an intermediate altitude. Given the long-distance attenuation, a slight change in communication paths via UAVs can lead to a significant gain in communication efficiency. UAVs can also provide flying buffers to ISTNs for opportunistic communications and handover operations. Indeed, recent experiments have shown that even a couple of ground or UAV relays can significantly improve the end-to-end throughput and latency

of an ISTN [13]. Furthermore, UAVs are more appealing for broadband applications since they operate at a much lower altitude than satellites, resulting in better signal reception power and lower latency. In addition, UAVs can provide images of higher resolution as compared to satellites. Besides, having a continuous communication opportunity between satellites is often impractical due to their lack of flying flexibility. It also means that most satellites spend a considerable amount of their time above oceans, deserts, or other largely unpopulated areas, where their bandwidth may be wasted. This leads to a lower area spectral efficiency (ASE) compared to the ASE of UAVs.

Overall, in NTNs and different from satellites, UAVs provide more freedom in network design while benefiting from flexible flying capability, which leads to greater performance in future communication services. With the aforementioned motivation, the overarching goal of this article is to identify the potential and challenges in UAV-based communications where UAVs can be dynamically linked together as nodes for communication purposes to create a so-called *flying ad hoc network (FANET)* [14,15]. Note that UAVs may also be known by other terms like unmanned aircraft systems (UAS) [16]. The key benefits of such networks are their flexibility, scalability, and robustness. In this respect, we revisit and aim to give a fresh look at FANETs in the context of NTNs. In the following, we explore the background of FANET in more detail.

### 1.3. Background of FANET

FANET is a sub-type of mobile ad hoc network (MANET) or vehicular ad hoc network (VANET) [17–20] as shown in Figure 2. However, while both MANET and VANET are mainly explored for ground-based devices on a 2-dimensional (2D) basis, aerial nodes may move freely and flexibly with a higher speed and a lower density in three dimensions (3D), posing new issues for network administration and operation. Besides, clear LoS propagation and environment-resilient communication are key enablers to establish highly efficient network topologies in FANETs [21]. We note that there are two main kinds of UAVs: fixed-wing drones and rotator-wing drones (either single- or multi-rotor) capable of vertical takeoff and landing [22]. The rotator-wing UAVs are more advantageous in terms of their high stability and flexible movements. In [23], a comprehensive characterization of distinctive UAVs was given in more detail.

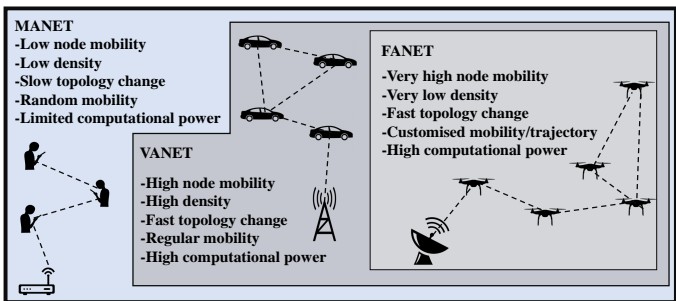

**Figure 2.** MANET, VANET, and FANET relationships. A detailed comparison can be found in [17,18,23].

It is noteworthy that the greater mobility of UAVs allows the network topology in FANET to quickly form a kind of fast time-varying networks [24]. Therefore, node mobility is one of the most critical challenges in designing FANETs, which is important for UAVs cooperation and collaboration [25]. Some mobility models have a predefined flight path and the map is updated after each modification [24,26], whilst others have no specific pattern [22,27,28]. Furthermore, when compared to MANET and VANET, the mobility of UAVs plays a vital role since their mobility, direction, and speed can change in considerably shorter time intervals, which might generate communication issues amongst UAVs [29]. This is perhaps FANET's most defining feature, enabling it to be suitable for trajectory control and optimization. Besides, power consumption [30,31], localization [32,33] and

radio propagation models [34,35] have been found as other critical factors with a huge impact on the performance of FANETs.

*Data packet routing* in such intermittent connected networks is another important topic where a huge corpus of research in FANETs is devoted to it. In [36–38], different routing protocols for FANETs are discussed and evaluated. Each protocol has distinct strengths and disadvantages, as well as suitability for specific situations, which makes it impossible to find one ideal routing protocol for all scenarios [39]. Hence, [19] classified the routing protocols in FANETs into five main categories as follows: (i) topology-based, (ii) position-based, (iii) clustering/hierarchical, (iv) swarm-based, and (v) *delay tolerant networks (DTN)*.

The first four categories, i.e., (i) to (iv), function well when the deployment area is not too large and most of the UAVs are in an LoS transmission range with each other, and the density of the UAVs is also relatively high. Within such a high density of nodes, communication from one UAV to another is reliable. However, we note that the sparsity of UAVs in FANETs is another intrinsic feature that has been identified for the intermittent connectivity problem. Therefore, the true issue arises when there is intermittent connections in a low density of UAVs. Indeed, [40] showed that in a social environment, such as a large urban region where various sites are separated by enormous geographical distances, there is a trade-off between UAV density and the cumulative energy consumption of the FANET. Therefore, [22] showed the first four categories of (i) to (iv) are relatively compromised. Moreover, [41,42] also explained that the the rise in the total number of UAVs in the network would be disadvantageous in terms of cost, message overhead, buffer overflow, and cumulative energy consumption. It was shown that in low UAV densities, **DTN routing protocols** are the only models that can send data packets with good performance like a high delivery ratio, despite the fact that the network is only sometimes linked [22]. Therefore, in our study, we mainly focus on the design criteria and architecture of DTN protocols in FANETs and call it *DTN-assisted FANET*.

### 1.4. Scope and Contributions

The primary purpose of this article is to deliver a discussion on NTN structure and its key components while narrowing the topic down from a satellite perspective into UAV-based NTN, i.e., FANET. After that, a thorough literature study on the most recent advances in FANET technology is presented. In this study, we aim to provide a high-level overview of the FANET and its most recent discoveries, which will include its benefits and drawbacks, as well as new research prospects for both current and future applications in 5G and 6G. We also detail the various specifications needed for FANETs and the key limitations that prevent its widespread deployment. Furthermore, we design a framework for a DTN-assisted FANET, while simultaneously covering well-known routing approaches in FANET. Applications of artificial intelligence (AI), machine learning (ML), and deep learning (DL) tools, in FANET are also discussed. With this as our main goal, the five important contributions that we made to this work are as follows:

- This paper presents detailed NTN with UAVs, i.e., FANET, including features, existing and emerging applications, and its constraints. It sheds light on the distinctions that exist between the NTN components and gives a comprehensive survey of 218 FANET-related papers.
- The holistic overview of most recent advancements in relation to the emerging FANET technology is provided in terms of communication standards, physical layer, UAV role management, trajectory optimization, and routing protocols.
- DTN-routing protocols are taken into specific consideration for FANET due to its nature of intermittent connectivity and a DTN-assisted FANET framework is described and evaluated.
- The applications of AI/ML/DL techniques in FANET are thoroughly discussed.
- Finally, we take into account FANET's potential and investigate its unique characteristics and advantages over existing approaches for dealing with challenging FANET

problems. We use this knowledge to foresee the paths future research will go and the obstacles that will need to be overcome to enable FANET in wireless networks.

We also emphasize the advantages of our survey while summarizing the characteristics of other important surveys of FANET in Table 1 to further elucidate the importance of this study. We highlight that, despite the previous important studies, our study primarily focuses on providing and bringing together a broad overview of NTN components, applications, communication standardization process, physical layer issues, network management, trajectory optimization, routing protocols, DTN-assisted FANET, and applications of ML in FANET.

**Table 1.** Summary of most recent Key Survey Papers in NTNs with UAVs, i.e., FANET.

| Ref. | Main Focus | NTN Components | App. | Commun. | Physical Layer | UAV Role Management & Trajectory | Routing | DTN-FANET Framework | AI/ML/DL |
|------|-----------|:---:|:---:|:---:|:---:|:---:|:---:|:---:|:---:|
| This Survey | FANET in NTN, state-of-the-art FANET, features of FANET, DTN-FANET perspective, UAVs trajectory/mobility | ✓ | ✓ | ✓ | ✓ | ✓ | ✓ | ✓ | ✓ |
| [17] | Difference between FANET/VANET/MANET, design criteria | x | ✓ | ✓ | ✓ | x | ✓ | x | x |
| [18] | Various routing protocols in FANET, features of FANET | x | ✓ | ✓ | x | x | ✓ | x | x |
| [23] | FANET architecture, mobility models, routing protocols | x | ✓ | ✓ | x | x | ✓ | x | x |
| [24] | Communication issues, routing, mobility, security | x | ✓ | x | x | ✓ | ✓ | x | x |
| [36] | Different routing protocols for FANET, architecture | x | x | ✓ | x | x | ✓ | x | x |
| [37] | Routing requirements of FANET, evaluation of existing routing protocols, UAV classification | x | ✓ | ✓ | x | x | ✓ | x | x |
| [39] | Existing routing protocols | x | x | x | x | x | ✓ | x | x |
| [43] | Power efficient protocols across physical, data link and network layers in FANET | x | x | ✓ | ✓ | x | ✓ | x | x |
| [44] | Routing demands, UAV functionalities, energy efficiency | x | x | x | x | ✓ | ✓ | x | x |
| [45] | Various cooperative approaches for FANET | x | ✓ | ✓ | x | x | x | x | x |
| [46] | Cluster-based routing protocols and their characteristics | x | x | x | x | x | ✓ | x | x |
| [38] | Mobility models and routing protocols | ✓ | x | ✓ | x | ✓ | ✓ | x | x |
| [47] | Joint trajectory and communication design for FANET | x | x | ✓ | x | ✓ | x | x | x |
| [48] | AI-based trajectory and routing protocols for FANET | x | x | x | x | ✓ | ✓ | x | ✓ |

### 1.5. Organization

In Figure 3, we see the overall outline of the article. Section II provides an overview of the preliminaries of NTNs while highlighting the significance of UAVs and FANETs. Moreover, the applications and use cases of FANETs are described in Section III. Next, Section IV presents an overview of the state-of-the-art FANET in terms of standards, physical layer advancements, network management, and routing protocols, with the major

focus on DTN routing protocols due to the nature of intermittent connections in FANET. Then, a DTN-assisted FANET framework is explained and evaluated in Section V. After that, Section VI explores the applications of AI/ML/DL in FANET. Section VII elaborates on the challenges and future research directions. Finally, Section VIII concludes the paper.

| I. | Introduction |
|----|----|
| II. | Structure of NTN and its Key Components |
| III. | Applications and Use Cases of FANETs |
| IV. | State-of-the-Art FANET |
| V. | Case Study: A DTN-assisted FANET Framework |
| VI. | Applications of Machine Learning in FANETs |
| VII. | Challenges and Future Research Directions |
| VIII. | Conclusions |

**Figure 3.** Organization of the paper.

## 2. Structure of NTN and its Key Components

This section introduces the key components of NTN along with its structure. It also explains the significance of UAV networks, i.e., FANET, in comparison with other NTN components.

In conjunction with terrestrial networks (such as cellular networks), a structure of NTN is illustrated in Figure 4 with geosynchronous Equatorial orbit (GEO), medium Earth orbit (MEO), and low Earth orbit (LEO) satellites. Furthermore, there are high altitude platforms (HAP) easing the communication in lower layers close to where airplanes fly. In the altitudes closer to the Earth, UAVs are supposed to be dominant flying objects facilitating real-time communication in both non-terrestrial and terrestrial networks. In Table 2, the technical specifications of these key NTN components are summarized while they are discussed in the following.

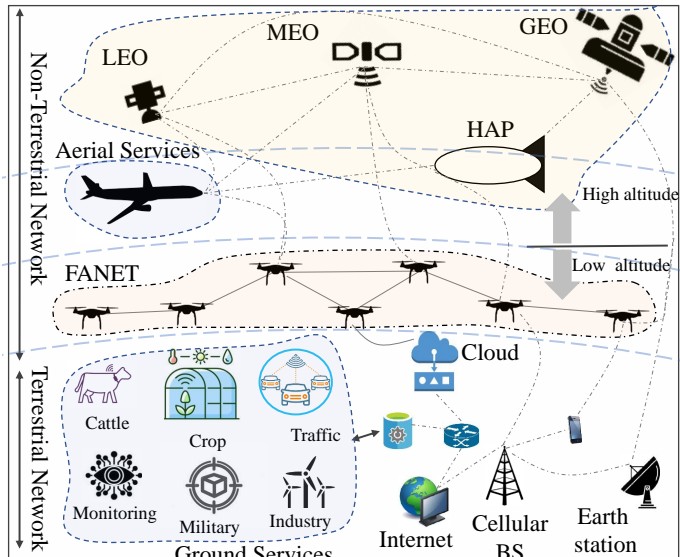

**Figure 4.** Structure of NTN and its key components. UAVs play a vital role to facilitate real-time and reliable communication in both non-terrestrial and terrestrial networks. These flying networks have a great potential to leverage current mobile communication networks to deliver advanced IoT services in the future.

**Table 2.** Key NTN components and their characteristics. UAVs are the lowest altitude platforms in the NTN that are advantageous in terms of cost, quick deployment, and flexibility in flight manoeuvres compared to other NTN components. RTT: Round trip time, BB: Boradband

| Type | Altitude [km] | Speed [km/s] | Uses | RTT [ms] | Coverage Ranking |
|---|---|---|---|---|---|
| GEO Sat. | 35–800 [$\times 10^3$] | ~3 | Relay, BB | ~500 | 1 |
| MEO Sat. | 2–35 [$\times 10^3$] | ~4.2 | Navigation, relay, backhauling | ~200 | 2 |
| LEO Sat. | 180–2000 | ~8 | High speed BB, imaging, backhauling | ~40 | 3 |
| HAP | ~ 20 | <0.3 | Fixed/mobile BB, Short/midterm backhauling. | 0.13–0.33 | 4 |
| UAVs | <0.5 | <0.07 | Communication, sensing, relay, high resolution imaging | ~1 | 5 |

1. **GEO Satellites:** GEO satellites are launched to orbit at an angular speed that is equivalent to that of Earth. Moreover, they are assigned to orbit along a route parallel to Earth's rotation (also referred to as geostationary or stabilized satellites because they appear stationary to the user on the ground), thus mostly delivering coverage to a defined and fixed area [49]. Deploying GEOs is typically very expensive mainly due to their launching costs. This is because they are allocated to orbit at altitudes higher than 35,000 km [50] (common altitude: 35,786 km). GEO satellites are mainly used for TV broadcasts and in some cases to relay communications between spacecrafts, including the space shuttle, the Hubble space telescope, and Earth-based control centers.

2. **MEO Satellites**: MEOs, also referred to as intermediate circular orbit (ICO), are satellites that orbit Earth between altitudes of 2000 km and 35,780 km (common altitude: 20,000 km). MEOs orbit Earth at faster angular speeds than GEO satellites due to their proximity to Earth. Indeed, as satellites are closer to Earth, the gravitational attraction becomes greater, and the satellites move faster [51]. Usually, it takes 2 to 24 h for one MEO satellite to complete a full orbit around Earth. They are mostly used for navigation systems, such as global positioning systems (GPS) [52].

3. **LEO Satellites**: LEO satellites are designed to orbit Earth at much lower altitudes, usually between 200 km and 2000 km. This enables LEO satellites to provide satellite services at relatively low delays, but at the expense of deploying more satellites [53]. However, since they are closer to Earth, they orbit much faster, i.e., (>25,000 km/h, and their orbit period varies over a range of 40–120 min. This means that each LEO experiences at least 12 and up to 36 morning and night periods in only 24 h [54]. Hence, a constellation of LEO satellites is proposed to compensate for and offer continuous, worldwide coverage for high-speed broadband communication as well as imaging and communication backhaul [53,55].

4. **HAPs**: Contrary to satellites and at lower altitudes of 17 to 50 km (stratospheric layer), HAPs (Additionally, known as high altitude aeronautical platforms (HAAPs))can be used to provide broadband communication services as well as broadcasting services by either unmanned airships, e.g., balloons, or airplanes [56]. HAP-based communication is suitable for large geographical areas where HAPs can move more freely and flexibly compared to satellites [57]. They are mainly powered by solar technology and non-polluting fuel cells.

5. **UAVs**: The use of UAVs is anticipated to be essential in 6G and beyond, thanks to their widespread and rising use in a variety of applications [58]. UAVs have a substantial advantage over other NTN components because of the free and flexible mobility of drones and their remarkable adaptability. In addition, they have several applications,

including expanding cellular coverage, agriculture, civil, military, industry, search and rescue, and fire monitoring, among others [19].

With the increasing adoption of the UAVs (The global commercial market of UAVs in 2020 was valued at USD 13.44 billion [59] mainly in industries such as media and entertainment, delivery and logistics, energy, agriculture, real estate, and construction, security and law enforcement, and so on. Because corporate use cases for UAVs have grown significantly in recent years, industry participants, such as UAV manufacturers and telecommunication industry partners, are continually inventing, testing, and enhancing solutions for UAVs network markets. From 2021 to 2028, the UAVs share market is expected to expand at a compound annual growth rate (CAGR) of 57.5% [59]. Therefore, based on what has been elaborated above and the significant growth expected in the future of FANETs, in the following, we narrow down the focus of this work to UAV networks and FANETs while the detailed investigation of other NTN components is deferred to our future work), their role in NTNs is becoming more vital since they are flying closer to the Earth compared to satellites in higher orbits, e.g., LEOs, MEOs, and GEOs or even HAPs. Therefore, they can provide shorter delays and reduce the round-trip-time (RTT) of NTN communications. Furthermore, UAVs can provide better spatial, temporal, or spectral resolution than satellites by providing complementary dimensions [60]. Moreover, UAVs have better ASE than other NTN components because other NTN components cover larger areas with a restricted finite bandwidth, limiting their use to largely unpopulated areas. In addition, since satellites are constantly moving, and the pathloss of signals is significantly high, communication and synchronization between satellites are far more challenging than in UAVs.

Overall, UAVs may outperform all other NTN components in some NTN use cases determined by 3rd generation partnership project (3GPP) such as (i) service reliability and (ii) ubiquity to provide NTN connectivity in the event of terrestrial network failure; and (iii) capacity to scale up service to meet peak demand for traffic from terrestrial networks [10,61]. Furthermore, the UAV networks can support services proposed by the international telecommunication union (ITU), which are enhanced mobile broadband (eMBB), massive machine type communications (mMTC), and ultra-reliable and low latency communications (URLLC) [10,61–63]. For these use cases, the actual value of UAVs is in the payloads they can carry and their ability to effectively establish networks of flying nodes, i.e., FANETs.

## 3. Applications and Use Cases of FANETs

There are various applications of FANETs including environmental monitoring and emergency communications, and UAV services as shown in Figure 5. In this section, we highlight four common real-life 5G/6G business use cases and the applications of FANET.

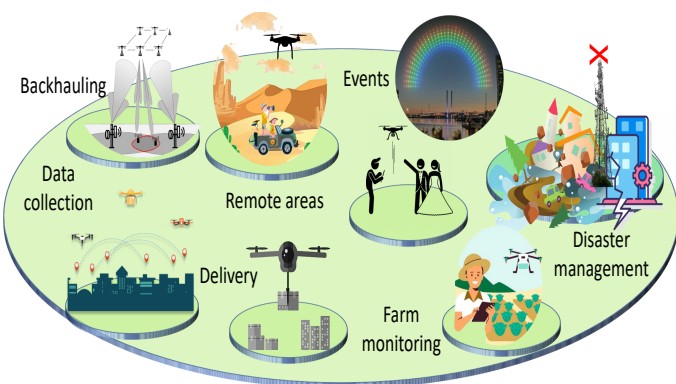

**Figure 5.** Applications and use cases of FANET.

### 3.1. Smart Farming

In order to increase production, smart farming aims to utilize IoT connectivity technologies. UAV-based IoT connectivity should enable farmers to remotely manage their crops using a variety of networked sensors and equipment in smart farming [64,65]. For example, a UAV can approach a passive sensor, like RFID sensors [66], on a farm and collect its data. Such integrated UAV-based data gathering of sensors is anticipated to significantly enhance smart farming in a number of ways. Data gathering in agriculture is fundamentally focused on enhancing and speeding up agricultural operations. The classification of UAV deployment in agriculture yields distinct applications of effective water management; crop monitoring; cattle and pest control; chemical level monitoring and management; etc. [67].

### 3.2. Emergency Situations

A possible application of a FANET is in disaster-struck areas (e.g., bush fire affected or flooded areas) where a swarm of unmanned UAVs is deployed to video capture and search the area for survival, or to inspect the damaged condition of the area, or form an NTN for use by survivors for a certain period. The area could be spread over tens or even hundreds of kilometers, and camera-equipped automated drones can capture footage of particular areas and stream (in almost real-time) the video to ground stations or to human rescuers in a helicopter, which can respond quickly as needed. Hence, the source of the video data is the drones themselves, and the video can be streamed from UAV to UAV (via the NTN) to reach a manned helicopter or ground stations. The NTN can also be used by survivors on the ground, who can connect to nearby UAVs (and so use the NTN) to send messages back to human rescuers, and even for survivors to be temporarily connected (to each other) where required. Moreover, more UAVs can be deployed if needed, and the NTN can be spread out further, reshaped, and resized accordingly as needed, providing flexible and dynamic on-demand customization of a network. Due to the limited battery power on each UAV, new UAVs can replace existing UAVs (nodes) in the NTN to maintain the NTN in certain configurations for the required time.

The general concept of deployment here is that an NTN (with particular functions (e.g., normal, thermal, or other specialized cameras) and characteristics such as certain levels of reliability, bandwidth, etc) can be spawned ad hoc as needed for a fixed period of time in order to serve particular areas, in contrast to satellite networks with (relatively expensive and limited bandwidth applications). Hence, such dynamic FANETs can complement satellite-based networks, where available.

Edge computing can be employed in such an application at different levels, from limited on-board (on-UAV) processing of some of the videos captured to processing in some more powerful UAVs or sending to nearby large helicopters (or airships) with more powerful computers (and more energy) for processing, and ground stations, before the processed video is sent to the cloud.

The above NTN using camera-equipped automated UAVs can also be used for non-disaster settings, e.g., an inspection of large-scale infrastructures such as power lines, pipelines, and roads to check their conditions, for security in large gatherings or events; and so on, where an ad hoc transient network that is flexible and reconfigurable would be useful. Such an NTN can be deployed for military purposes and for outdoor expeditions as well, to support operations and explorations in difficult-to-reach areas for a certain period of time. Note that an NTN can be made secured forming a private network for use over an area for a fixed period of time, e.g., the police or fire brigade can deploy and use such an NTN to support operations on an ad hoc basis.

### 3.3. FANETs for Events

We have seen several fascinating deployments of a fleet of UAVs in events to create drone light shows and provide 3D imagery (e.g., each drone as a "pixel") at important events. (For example, see https://skymagic.show/ and https://agb.events/drone-skyshow/ [last accessed on 15 July 2022]) However, this is only one way that a fleet of drones can

support events. The ability to deploy or spawn a FANET over a large event, e.g., during Olympics events, a large festival in a regional area, or a large rally such as a New Year's Celebration event, opens up new possibilities for improving people's experiences of such events, and possibly new commercial event-drone services.

For example, a FANET could support connectivity services for people on the ground in an event over areas under-served by 4G/5G networks (e.g., in a regional town). Expensive infrastructure or base stations cannot be deployed overnight for an event that would just take place over several hours, or even over a few days or weeks - instead, a FANET can be used to provide connectivity for the period of the event according to requirements. A higher density of more powerful UAVs could provide higher bandwidth and higher reliability connectivity if this is needed (e.g., high-quality video captures, 4K to 12K ultra HD) but an event that does not need it might only use a smaller fleet of UAVs that can support lower connectivity. Maintenance of such connectivity may involve UAVs replacing current UAVs to maintain the FANET connectivity services. UAVs can even be multi-functional, providing a light show at certain times and connectivity at others, or both at the same time, if their energy capacity allows. Such UAVs can complement any other network connectivity made available (e.g., WiFi stations on the ground). Such a FANET can also provide on-demand ad hoc fly-in and fly-out edge computing services, with powerful drones acting as servers [68].

*3.4. Cooperative Actions and UAV Air Traffic Management*

Not only to provide network connectivity services to people or user devices but a fleet of UAVs might also be used for "physically connecting a region" in a way that is optimized and cooperative. The FANET in this case might be used by the UAVs themselves to cooperate while performing their tasks, e.g., maintaining a physical delivery network (e.g., delivering supplies and goods) among disparate sites over a region, or to cooperate in optimizing their flight routes, or cooperate in coping with unforeseen events and accidents, or even to avoid UAV-to-UAV collisions. For example, UAVs can cooperate in path planning and with mesh-like connectivity to allow UAV-to-UAV connectivity over ranges beyond the limits of direct UAV-to-UAV transmission to provide advanced anticipative collision warning/avoidance and dynamic evasive maneuvers and re-routing well ahead of time. In future air spaces, we can envision UAVs of different sizes, uses, and capacities (e.g., small UAVs with cameras for surveillance and hobbyists UAVs, small to medium-sized UAVs for goods delivery or logistics, and larger UAVs for people transport) occupying the same or similar air space at higher densities, e.g., over urban areas [69] (where we have the notion of invisible urban "*highways in the sky*" (For example, see the upcoming project on creating highways for drones in the UK: https://www.youtube.com/watch?v=57o7JmarqTs [last accessed on 25 July 2022])), where UAV-to-UAV connectivity can help avoid UAV-to-UAV collisions (in a way similar to, but more complex, than the "2D" ground vehicle-to-vehicle communications being used to help avoid vehicle-to-vehicle collisions and facilitate vehicle-to-vehicle cooperation on a large scale).

## 4. State-of-the-Art FANET

In this section, we provide the state-of-the-art and key latest findings of FANETs in terms of (i) communications standards, (ii) physical layer aspects, (iii) role-based connectivity and trajectory management, and (iv) routing protocols.

*4.1. Standards for FANETs*

In the last decade, there has been an extensive effort to envision the FANET and industrialize it. As a result of the benefits provided by FANET and to take economic advantage of it, 3GPP approved to support FANET integration into the 5G ecosystem [61,70]. In Release 16 [62], 3GPP accepted the first study item on NR to support FANET. The goal was to investigate channel models (propagation conditions and mobility), establish deployment scenarios and related system parameters, and identify any important effect areas that may

require more research. Similarly, the Institute of Electrical and Electronics Engineers (IEEE) and ITU have made some efforts towards FANET standardization. In IEEE P1936.1 [71], IEEE proposed a standard framework for FANET applications. In IEEE P1939.1 [72], structuring of UAV operations in FANET was standardized. In IEEE P1920.1 [73], communications and networking standards were clarified. ITU also shed light on the FANET architectural standardization in ITU-T Y.UAV.arch [74] and ITU-T F.749.10 [75] and provided communication requirements for UAV-based services.

Following these standardization pathways, in [36], communication protocols were considered a general issue of FANETs. In [15], adaptive communication protocols were proposed based on position-prediction and reinforcement self-learning routing techniques. The beauty of this work was the capability of developing communications with respect to the modifications happening in the FANET topology. In [76], a joint sensing and communication protocol [77] was proposed that enables the FANET to be integrated within the cellular networks. It was also proposed to use non-orthogonal multiple access (NOMA) where several UAVs/nodes can access radio resources non-orthogonally to reduce the latency and improve the spectral efficiency. Moreover, [58] indicated three main aspects of FANETs to support UAVs operations in different applications. They were: (i) dependable 3D wireless access with integrated cellular networks and FANETs, (ii) AI at the edge UAVs for signal/image processing and smart decision makings, and (iii) effective control and inter-UAV communications for cooperative activities. In addition, an integration between FANET and the cellular networks was proposed to provide strong connection, dependability, security, and safety to UAVs by using their sophisticated features and UAS traffic management (UTM) (UTM's goal is to provide the reliable transportation of products and services through UAS/FANET [78]) supports. On the other hand, the studies in [21,67,79,80] proposed aeronautical channel models for air-to-air (A2A), air-to-ground (A2G) and ground-to-air (G2A) communications in cellular FANETs, as shown in Figure 6. Aly and Vuk in [81] further investigated communication standards for FANETs in terms of remote identification of UAVs, command and control communications, radio access network (RAN) support, and various solutions to the issue of RF interference during cellular UAVs integration. They also discussed FANET node registration and wireless service requirements.

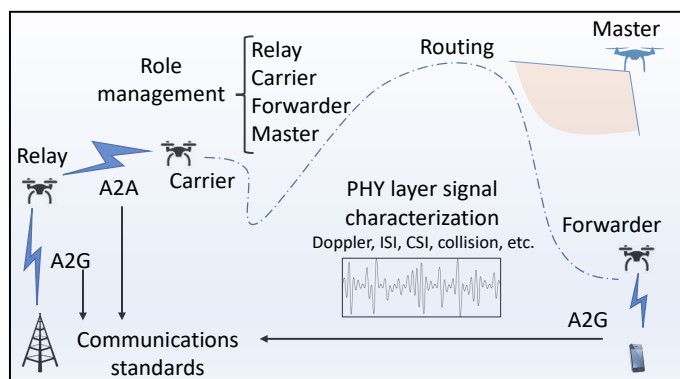

**Figure 6.** Advancements of FANET are mainly in communication standards, physical layer characterization of data transfers, role & trajectory management of UAVs, and data packet routing. Communications can be classified into three classes of A2A, A2G, and G2A. Based on the network design, each UAV may have four main roles of relay, carrier, forwarder, master. The Master UAV is responsible for the network management as a core.

To conclude, although a wide variety of short- and long-range technologies, given in [61,62,71–75,82], can be helpful in FANETs for specific applications, the communication framework standardization of FANETs is an ongoing direction with an essential need for significant experimentation platforms to bridge the gap between basic research, stan-

dardization, optimization, and deployment as well as existing and future use cases and research difficulties.

In the following, we further elaborate on the physical layer advancements of FANETs, highlighting their significance in the network performance.

*4.2. Physical Layer Advancements in FANETs*

Full optimization and deployment of a UAV-based communication network would not be accomplished without the profound investigation of suitable physical layer protocols. At the physical layer where the nodes perform the data transmissions in different forms of waveforms and numerology designs [83], the selection of an appropriate interface, like those explained in [82], affects data rate, coverage range, reliability, throughput, and other system requirements. Nevertheless, the investigation of the physical layer in FANETs includes various topics such as channel state information (CSI), Doppler effect, inter-symbol interference, packet collisions, etc.

Herein, a hierarchical model in [35] showed that the physical layer has a significant impact on the efficiency of FANET networks. It detailed signal propagation of different bands, including sub-6GHz and mmWave bands in different levels of the atmosphere; signal creation and conversion in FANET transceivers; and signal conversion in antennas. Furthermore, attenuation and inter-symbol interference (ISI) were considered in that model by the proposal of inertial frequency-dependent links and multipath channels. Besides, the authors in [84] analyzed the mmWave UAV-assisted networks with the vision of providing wireless mobile access. Another model in [85] proposed a cross-layer of physical- and network-layers for anti-jamming routing protocol in FANETS. It used cognitive radio concept along with taking the physical layer link quality parameters into account for routing decisions. Indeed, it showed that the use of cognitive radio in FANETs improves spectrum efficiency while allowing UAVs and ground-based cellular users to cohabit in the same frequency range. Furthermore, in contrast to conventional networks, UAV networks are capable of having more transparent mobility models, which enables more precise forecasting of time-varying interference. The authors in [86] proposed a technique to predict the level of interference that is applicable to such UAV networks. In addition, in [87], a general framework was proposed to forecast the interference behavior via analysis of the mean value and moment-generating function of the interference prediction. Moreover, the authors in [88] designed parameters for adaptation of the physical layer for data transmission in self-organizing FANETs. Furthermore, a communication interface manager was proposed in [89] for improving the performance of heterogeneous FANETs. It dynamically defined the optimal means of transmitting data based on real-time analysis of the state of the wireless medium. In [90], the physical layer in 5G new radio (NR) was modified to piggyback the FANET. It assessed the waveform and a scalable numerology for both frequency division duplex (FDD) and time division duplex (TDD), and a multi-antenna transmission and beamforming.

Physical layer security for UAVs in FANETs, like models proposed in [91–93], is another important topic with great potential to exploit the randomness nature of the wireless channels for secure communications. Practical UAV channel modeling, position acquisition [94], pilot contamination [95], limited buffer size and resources [96,97], and NOMA [98–100] are other important aspects of the physical layer that affect the system performance in FANETs significantly. For instance, the study in [84] depicted the performance of NOMA on the FANETs in terms of power efficiency, data-rate improvement, and radio resource allocation.

Overall, within the physical layer realm of FANET, there is still more work to be done to have a general framework. Having a full framework will further help to enhance the system modeling, which results in more reliable data processing in higher layers.

### 4.3. Role-Based Connectivity and Trajectory Management

Management of the FANET is another crucial task for efficient networking. The general objective is to minimize the intermittent disconnectivity between the nodes and therefore reduce the latency. For example, among the nodes in a FANET, some nodes may have a higher priority to be in the connectivity loop or even a node can move in a way to prevent the FANET from splitting up into multiple disconnected clusters. This requests precise role management of the UAVs in the FANET where three management options can be available as:

(1) a series of ground base-stations to command and control the UAVs,
(2) some of the UAVs work as master UAVs and control others, or
(3) each UAV learns how to deal with the network variations by using edge technologies such as AI and ML tools.

Depending on a scenario or a specific application, the optimal solution might be a hybrid model where all these three options become accessible. In this regard, [21] proposed an agent-based machine for connectivity management of UAVs by introducing four states for each UAV. Using this agent-based machine, each UAV can manage its role based on the information it gets from the nearby environment. However, this model is suitable for a swarm of UAVs when the nodes are somehow connected to each other at all times. In [101], a modular relay positioning method was proposed to manage the FANET to maintain the connectivity of the UAVs during missions with the least number of nodes. Likewise, in [102], a dynamic relay selection and positioning for FANETs was proposed. Moreover, in [64], a new model of data-aided management of FANET was proposed such that sensing and communication occur on demand in response to a specific query. The model was to collect measurements from massive passive sensors while adaptively minimizing the entropy gap and scheduling UAVs to fly over sensors.

Path planning and management is an important topic for the mobility of FANET nodes. Although in many literature surveys, the mobility of UAvs treated like nodes in MANETs or VANETs, the mobility of FANETs must be more deterministic based on the scenario specifications since, unlike nodes in MANETs/VANETs, the nodes in FANETs can freely and flexibly fly in a 3D space. Therefore, it demands a kind of application-oriented trajectory optimization for FANETs. In [103], a model was proposed to manage the radio resource among UAVs, and to optimize the trajectories of UAVs in the network by utilizing the tools of reinforcement learning. Hence, [104,105] proposed altitude optimization models while [47,48,106] focused on general trajectory optimization approaches for UAVs in FANETs. In general, while UAV trajectory design and route planning are crucial to improving FANET performance, the specifics of these two factors vary greatly depending on the needs of each application. As a result, different application-oriented trajectory optimizations have been proposed in the existing literature, like the models in [107–112].

All in all, efficient role management and trajectory optimization of UAVs is a real-time task which requires the ability to adapt to network changes and act accordingly. This demands for more application-oriented standardization and scalabel path planning protocols.

### 4.4. Routing Protocols in FANETs

As shown in Table 1, almost all the recent key surveys discussed the importance of packet routing protocols [14,19,36,38,67,113–116]. In general, a classification of routing protocols divides the available protocols into four groups of static, proactive, reactive, and hybrid [20]. The static models require a fixed/static routing table, and this routing table is not being updated during the service period. The proactive models periodically update the routing table. On the other hand, reactive models find routes for communications on demand, while hybrid models mix proactive and reactive approaches.

On the other hand, another classification model classifies the available routing protocols into seven categories, as shown in Figure 7. This classification is also admitted in [23]. Topology-based models (given in Appendix A) rely on source-to-destination information recorded in a routing table. The routing table may vary during the time since the position

of the nodes changes. However, the position-based models (given in Appendix B) do not require a routing table, and a decision is made at each node taking into account the current locations of nearby UAVs and intended destinations. The secure-based models use a security mechanism like network coding strategies [117] to assure the safe transition of messages through intermediate UAVs. Heterogeneous routing models expand the packet route via different interfaces used in different nodes [118]. It can facilitate the use of existing infrastructure in accomplishing specific tasks. The swarm-based model [119] is a fully connected FANET (like a murmuration of birds), usually used for quick data collection purposes since it has the ability to collect a huge amount of data in a short time interval. Clustering protocols [120] usually have a cluster head for each group of UAVs. The cluster head controls the features of each cluster and manages the routing of packets from one cluster to another. Eventually, energy-based routing [121] prioritizes the energy consumption of FANET for routing a packet.

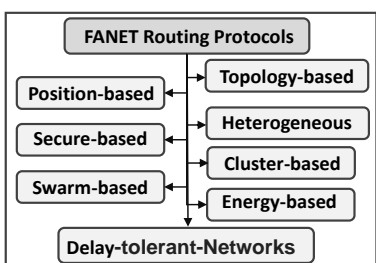

**Figure 7.** Classification of FANET routing protocols.

The aforementioned protocols mainly consider the high density of UAVs flying in each other's range. Indeed, they somehow underestimate the challenges of maintaining steady and dependable connections between UAVs in lower density scenarios, whose connectivity may not be consistent due to the unstable nature of wireless communications [122]. Moreover, they usually suffer from large message overhead and buffer overflow during the flooding of messages and network information. The huge cumulative energy consumption of these protocols is also a drawback for high-density FANET. As a result, and as shown in Figure 2, the FANETs are supposed to be very low-density networks with a limited number of UAVs in a large area. Consequently, there will be constant outages, partitions, and topology changes. Hence, DTN routing protocols, shown in Figure 8, can be used in a way that store-carry-and-forward (SCF) is used by each UAV to deal with sporadic network connections. DTN routing protocols have a very low overhead at the expense of latency increment since no control messages are transmitted. Among the DTN routing protocols shown in Figure 8, LAROD is the most widespread protocol used in FANETs based on [19]. In the following, a DTN-assisted FANET based on LAROD [123] is fully reviewed and evaluated with respect to alternative routeing protocols in the DTN.

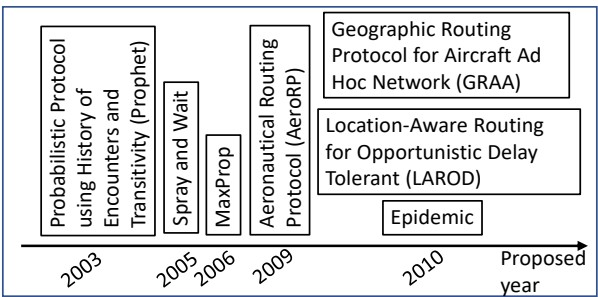

**Figure 8.** Seven well-known existing DTN routing protocols to be used in FANETs. Detailed descriptions of these models can be found in [123–129].

### 5. Case Study: A DTN-assisted FANET Framework

In this section, we profit from the LAROD technique [123] to explain a general DTN-assisted FANET framework. Below, we explain it in three stages and then compare it with other DTN routing protocols in a low-density FANET environment.

1. Suppose a downlink scenario, shown in Figure 9, with *N* nodes where a source node, e.g., a gNB, broadcasts a packet to other neighbor forwarder nodes, e.g., UAVs, in its coverage range. The objective is that the best UAV gets selected as the best forwarder based on its location and re-broadcasts the packet to other forwarder nodes in the next hops, i.e., UAVs. This process is to continue till the packet travels the hops and reaches the destination node, i.e., the user.

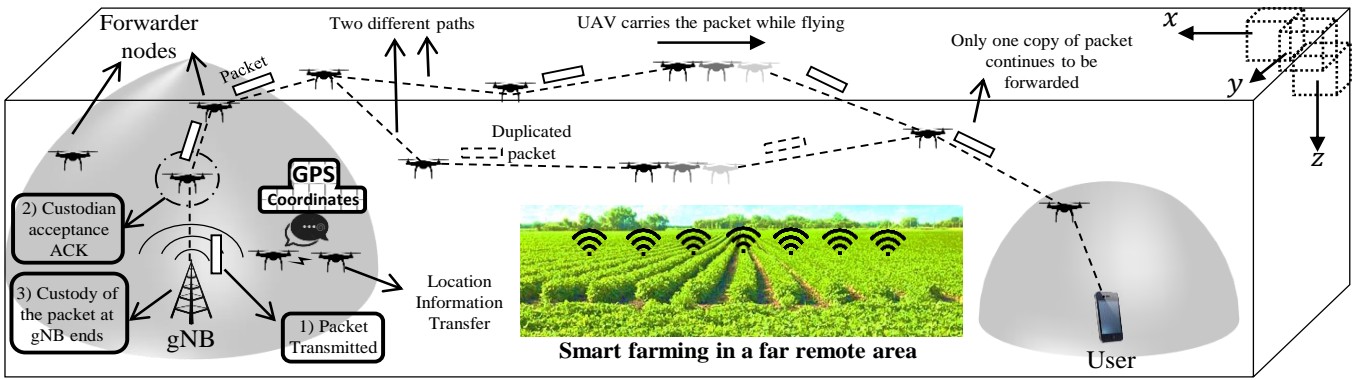

**Figure 9.** Proposed DTN-assisted FANET model framework. gNB: ground new base station.

For the purpose of best forwarder selection in each hop, a proactive flooding-based location service (FLS) is required to enable each UAV to continuously disseminate a current map of the network's neighborhood. For this purpose, each UAV has a GPS module and an inertial measurement unit (IMU) (The IMU calibration using GPS signal enables quicker delivery of UAV position coordinates) to exchange location information with other UAVs at rendezvous points. A portion of the buffer in each UAV is also dedicated to the location information of the whole network. Then, when two UAVs get together, they share their network's coordinates, and the most recent information will be kept while the older one will be discarded. This happens proactively to have a kind of socially aware network. The issue with this system-wide dissemination of location information seems to be the use of a lot of system resources. However, the realistic number of UAVs in current FANET applications hardly exceeds 20–30 UAVs; because each UAV has a good ASE and flying range compared to ground nodes, making this number of UAVs enough to cover a wide area of ground users. Therefore, the buffersize for the coordinates of this limited number of UAVs is legitimate. On the other hand, because of the intermittent connectivity between the UAVs, the location data of nearby UAVs is more up-to-date while that of UAVs further away might be outdated. However, [123] proved that even the outdated location data of far UAVs can be used for the routing of packets as the precision of the location information improves as the data packet proceeds towards the destination node.

2. At stage two, the best forwarder should be selected. Let us assume the broadcasting node is the current custodian and the next forwarder will be the next custodian. Hence, all UAVs that received the packet from the current custodian are tentative custodians. Based on the Bundle protocol, RFC5050 [130], in DTN systems (Bundle protocol is a custody-based retransmission DTN protocol created for shaky and intermittent networks. To communicate, it bundles together blocks of data and sends them all at once, using the SCF method), the tentative custodians activate their delay timers on the arrival of the packet, and if each custodian's timer runs out first, it becomes the next custodian, i.e., the best forwarder. Once its timer delay runs out, it stores

the packet in its buffer and broadcasts a custody acceptance acknowledgment (ACK). Then, the current and other tentative custodians that hear this ACK discard the packet. In this modeling, there would be two scenarios of hidden terminal problems: (1) if the current custodian does not hear the ACK it repeats the transmission after a fixed period of time, and (2) if any of the tentative nodes does not hear the ACK, it sends its own ACK and becomes a parallel custodian for that packet. As a result of both of these scenarios, there will be a chance of duplicated packet transmission through multipath directions. It is also possible that the duplicated packet from two separate paths reaches a single custodian and, from there, only one copy of it continues to be forwarded. The duplication of a packet increases the load on the network, so to end this, there is a time-to-live (TTL) period for each packet. After this period, the packet would be discarded by its custodian.

3. Steps one and two continue between the UAVs hop-by-hop till the packet reaches the destination node at the application layer. The destination node then broadcasts an ACK indicating the packet has reached the destination. The UAVs that receive the ACK store it in their buffer and exchange it with other UAVs carrying the packet till the packet TTL runs out. As a result, all other custodians carrying the packet are notified of the delivery and discard the packet from their buffer.

Figures 10 and 11 show performance comparisons between this LAROD-based model and other DTN-assisted FANET techniques proposed in [22]. Simulation is carried out using the opportunistic network environment (THE ONE) simulator with simulation parameters given in Table 3. Random Waypoint mobility is considered. The rest of the parameters are the same as those in [22] unless otherwise specified. The UAV nodes are supposed to be small, e.g., DJI mini2 $20 \times 20$ cm$^2$, and their minimum distance cannot be less than 1 m. We can see that the LAROD-based model slightly outperforms other DTN-assisted FANET models in terms of delivery ratio. We note that although the MaxProp model slightly outperforms the LAROD-based model as shown in Figure 10, MaxProp's latency is much higher than the LAROD-based approach as shown in Figure 11. Figure 12 shows the average buffer time for each delivered message, and we can see that the MxProp routing has the most efficient buffer time since it can prioritize the packet queuing with additional processing at the nodes. On the other hand, the Epidemic routing has the worst performance in terms of average buffer time since it broadcasts packets to all nodes and UAVs keep the packets in their buffer.

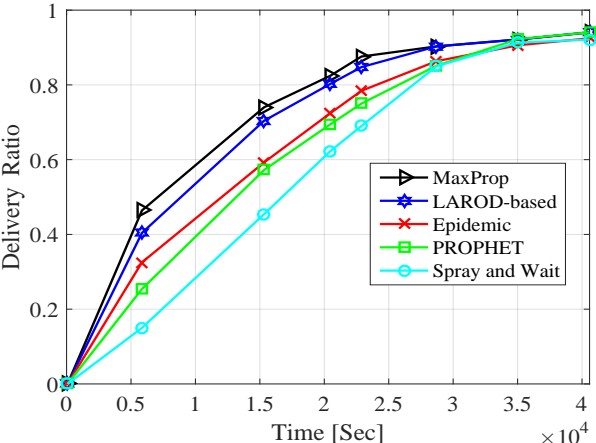

**Figure 10.** Delivery ratio comparison of LAROD-based model with a few well-known DTN routing protocols. The number of UAVs is set six.

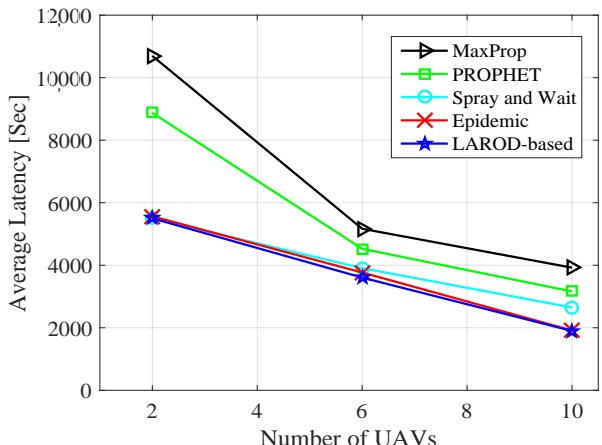

**Figure 11.** Latency comparison of LAROD-based model and a few well-known DTN routing protocols.

**Table 3.** System numerical parameters.

| System Parameters | Corresponding Value |
|---|---|
| Number of UAVs | 2, 6, 10 |
| Speed of UAVs | Varies between 1 m/s to 5 m/s |
| Speed of mobile user | 1 m/s |
| Interface model | Bluetooth (IEEE 802.15.1) and WiFi (IEEE 802.11b/g/n) |
| Transmit speed | 250 kBps |
| Message size | 250 KB |
| Transmit range | Bluetooth: 20 m, WiFi: 100 m |
| Buffer Size | 10 GB |
| Mobility model | Random Waypoint |
| Message TTL | 300 min |
| Simulation running time | 12 h |

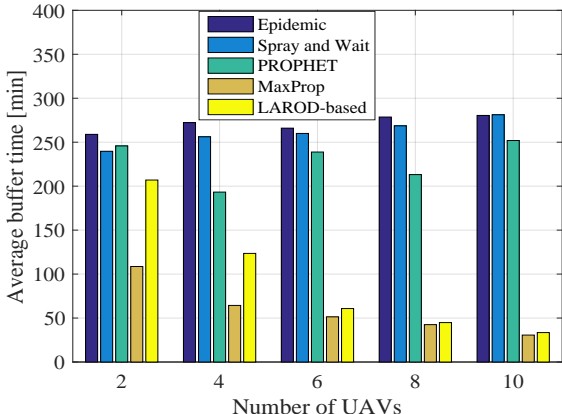

**Figure 12.** Average buffer time of a delivered message $\leq$ TTL = 300 min.

Figure 13 shows the average number of hops in each protocol. We can see that an increase in the number of UAVs slightly increases the average number of hops. LAROD-based model is being more affected by the number of UAVs compared to the other models. Figure 14 demonstrates the overhead ratio as the difference between relayed and delivered messages on the basis of delivered messages [18]. We note that the Spray and Wait protocol has the lowest overhead, while the Epidemic and PROPHET models have the largest overheads.

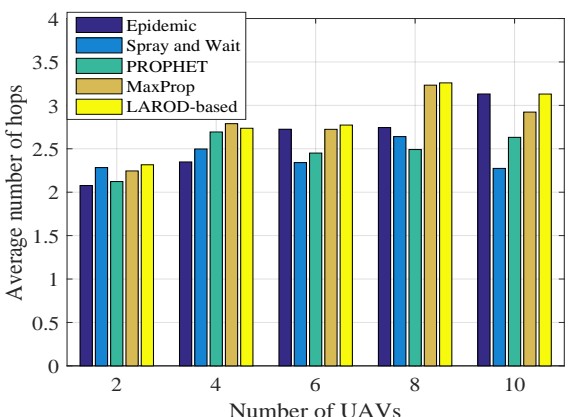

**Figure 13.** Average number of hops (node exchange) in each protocol.

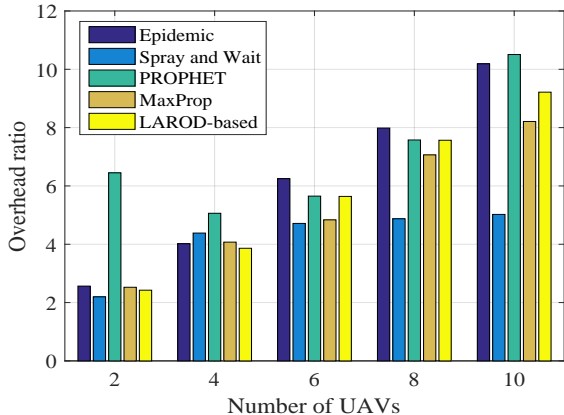

**Figure 14.** Overhead ratio of different protocols.

## 6. Applications of Machine Learning in FANET

Fundamentally, FANETs are expected to facilitate end-to-end data delivery using multi-hop (Single-hop transmission is omitted in this study since it is well-addressed in the literature [131,132]. Hence, they rely on multiple UAVs that are constantly working together in conjunction with the ability to create a system that is far more powerful than any one UAV alone.) transmissions. Therefore, each UAV or node functions by relaying information between nearby drones. When the number of hops increases (and the network becomes larger), the complexity of the system significantly increases. Accordingly, this leads to difficulties in network management and operations due to elevated interference levels, challenging spectrum allocation, demanding trajectory and mobility management, reduced localization accuracy, and more complex energy and multiple access/routing management.

The utilization of ML techniques has been shown to provide low-complexity solutions in wireless networks [133–135]. Due to its technology agnostic nature and adaptability, ML is quite flexible and can be used across the different layers of wireless networks [136]. Most notably, AI and ML have led to significant improvements in (i) the physical layer by enhancing digital signal detection [137], (ii) the medium access layer through cognitive radio and spectrum access [138], and (iii) the network layer using network management and optimization [139]. Hence, AI can potentially provide FANETs with the tools necessary in order to overcome some of the different challenges.

In particular, reinforcement learning (RL) has emerged as a prominent solution to tackle some of the different challenges faced in FANETs [140]. Most notably, Q-learning and deep Q-learning (DQL) are currently the most explored in the literature for FANETs to address their challenges. Owing to its ability to perceive, interpret, and learn from its environment, these techniques enable FANETs to adaptively collect data and optimize network performance. Both Q-learning and DQL rely on states and rewards to understand

and learn the environment and enhance their actions through weighted trial-and-error to optimize the network's overall behavior [140]. Recent works that deploys RL have tackled many avenues for FANETs. In this section, we focus on three main RL-enabled domains; (i) routing, (ii) resource allocation, and (iii) network security.

### 6.1. Routing

RL provides the nodes with the capability to learn from their environment in order to autonomously and adaptively select the most suited next-hop node in packet routing. This has motivated researchers to develop routing techniques based on Q-learning. The role of RL for routing in FANETs was thoroughly investigated in [141] by assessing over 60 routing protocols that were proposed since the 1990s. It recommended RL as the best solution with reasonable overhead in terms of control packets, memory, and computation. In [142], the authors introduced a routing technique dubbed as QGeo which utilizes Q-learning. This methodology aims to reduce the routing overhead while maintaining a higher packet success delivery in comparison to other methods that do not rely on Q-learning. Moreover, the authors in [143] proposed a modified version of QGeo, where they enabled the system to learn the reward function of the Q-learning function in real-time through the inverse RL mechanism. The authors demonstrated an enhancement to the shortcomings of QGeo through real-time learning.

Another significant Q-learning-based routing technique that deploys the multi-objective optimization routing protocol, dubbed as QMR was introduced in [144]. QMR enables the adaptive learning of the network to its dynamic environment. In [145], the authors proposed a routing scheme called Q-FANET which uses a specialized version of the Q-Learning algorithm designed to cut down on delivery times in highly mobile networks. The authors also demonstrated enhanced performance in terms of packet success ratio, delay, and jitter when compared to the other RL-assisted routing techniques.

There are other algorithms that incorporate RL with other tools that have also been explored for FANETs. The authors in [146] proposed a routing protocol using fuzzy logic and Q-learning to enable routing path selection based on reducing the number of total hops and minimizing the delivery time. In this case, Q-learning is used to support the output obtained from fuzzy logic through a point system for evaluating potential routes. In addition, the authors in [15] relied on the use of Markov decision processes to propose an adaptive hybrid communication protocol based on RL.

While the current mechanisms have been shown to provide a major enhancement in the network routing performance, the complexity of the system becomes very high with more inputs. The primary reason for this is the centralized nature of the proposed techniques, where all the data is communicated back to the server, which is responsible for the training, rewarding, and action-taking. Distributed learning could provide merit in large networks, which can enable the different nodes to adjust their parameters in real-time [147,148].

### 6.2. Resource Allocation

In order to achieve seamless service while maintaining costs and energy consumption constraints, FANETs require real-time, autonomous, and dynamic resource allocation techniques. This is even more crucial in FANETs in comparison to static terrestrial links since each UAV requires the selection of a flight trajectory, which ensures that the UAV is maintaining its coverage, energy consumption, and avoiding obstacles (Obstacles could be static, such as high-rise buildings in urban environments, or even dynamic, such as other UAVs or departing/landing airplanes that are at the same altitude). Other issues include the allocation of UAVs to certain geographical regions or the redirection of some UAVs to cover for others that might require charging. Traditionally, logistic regression and support vector machine (SVM) have been commonly applied to UAV resource allocation problems [149]. In recent years, neural network-based and deep RL-assisted methods have become more popular to cope with the complex and dynamic nature of FANETs [140].

In [150], a ML algorithm was proposed focusing on the integration of UAVs into terrestrial networks to find efficient resource allocation along with the optimized 3D locations of UAVs to maximize the sum logarithmic rate of the users.

In [151], the authors suggested a model for optimizing the power transmission, trajectory design, and user association for a multi-UAV setting. In this setting, a central UAVs or a base station chooses an action for every time slot following the Q-learning method. In order to attain the required, the battery state, the state of the channel, and maximum transmission state were considered. Further, the researchers suggested that simple use of Q-learning is not advisable due to the high dimensionality problem that it presents. To further elaborate, due to the dynamic nature of UAVs, it is possible that thousands of states exist. Then if the values are stored in the Q-matrix table, there would be very large and insufficient samples to traverse every state of the UAV, often leading to the algorithm failing. As a result, the authors suggested combining Q-learning and Deep Neural Network (DNN). Instead of calculating the Q-value for each state, Deep Q-Network (DQN) can be used to estimate the value function for Q-learning.

In [152], resource management was studied in UAV-enabled long-term evolution (LTE) settings over licensed and unlicensed bands. The major aim was to optimize users' QoS while fulfilling each user's latency requirement. Instead of using conventional echo state networks (ESNs) for this, the authors suggested using ESN with leaky integrator neurons. This was driven by the algorithm's capability to dynamically update the ESN's status for users and achieve the optimal user QoS. Through simulation, it was shown that this method outperformed Q-learning with LTE-U and Q-learning with LTE in terms of performance.

Due to UAV privacy concerns and their computational limitations, distributed computing has recently gained attention. One such method for networked computing is federated learning (FL), which was developed by Google [153] in 2016. In FL, users do not exchange data, and models are locally trained. The base station conducts an aggregation function on the model parameters and generates a global model, which is then delivered back to the users, using the local models as input. In [154], the authors presented an asynchronous FL framework for resource management. Furthermore, an asynchronous advantage actor-critic (A3C) algorithm was implemented for device selection, UAV placement, and efficient resource management. The authors emphasized the usage of deep RL algorithms, such as DQN, which utilize experience replay for effective learning. However, leveraging experience replay necessitates having enough memory and processing capability to ensure more accurate learning. As a result, using the A3C algorithm circumvents these restrictions and may be deemed "better" in scenarios where memory size and computational resources are limited, which is frequently the case. Simulating this scenario concluded that this framework outperformed the existing solutions for resource management in terms of efficiency and learning accuracy.

In addition, the application of FL for power allocation and scheduling policy over a swarm of UAVs was suggested in [155] using the derived knowledge from transmission delay and antenna angle variations on convergence. The simulation results showed that FL convergence was more effective than a baseline architecture.

*6.3. Network Security*

The security of each node is paramount to ensure the security of the network. As a result, as the number of nodes in any wireless network increases, it becomes more vulnerable to jamming, and network breaches, among many other threats. Moreover, it becomes more challenging to monitor and protect each node individually due to the centralized nature of the operation. Therefore, network security is another important domain in FANETs where AI techniques, specifically distributed and/or FL, can be employed to protect the network.

The FL has shown great potential in FANETs to detect and combat jamming by demonstrating its capability as the number of nodes increases. For instance, the authors in [156] proposed an FL architecture to detect jamming with a prioritization technique, laying the foundation for FL in jamming detection. In addition, they developed an adaptive

anti-jamming methodology that is based on federated Q-learning in [157]. Another work also proposed the use of federated Q-learning networks to achieve a frequency hopping strategy to combat periodic jamming [158].

In addition, FL is capable of incorporating many algorithms that can better secure the communication channel between the different nodes and the cloud [159], which can be deployed in FANETs to combat eavesdropping attacks, while AI-enabled solutions for security purposes have demonstrated potential in wireless systems, there are still few solutions that are suitable for networks such as FANETs, thus they require more investigation.

## 7. Challenges and Future Research Directions

In this section, we discuss challenges and future research directions of FANETs with UAVs.

### 7.1. Infrastructure-Aided Control Plane

A FANET without terrestrial infrastructure needs to maintain itself in a decentralized manner as conventional ad hoc networks. On the other hand, when a FANET is to be operated with terrestrial networks, it can take advantage of terrestrial infrastructure. To this end, gNBs can help. For example, as an extreme case, the coverage of gNBs can be extended to support all UAVs in a FANET to exchange information in the control plane, as shown in Figure 15, which might be possible as the data rate for the control plane can be relatively low. In this case, any UAV directly communicates with a gNB to exchange control information at a low rate in the control plane that uses a different network from the FANET for the data plane, where data packets are transmitted through UAVs.

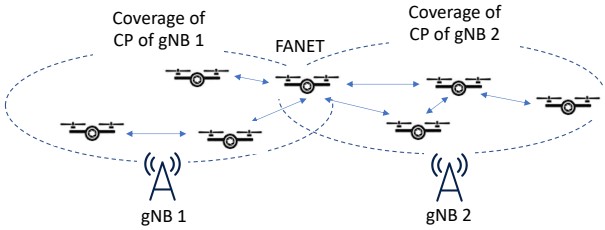

**Figure 15.** Coverage of control plane of gNBs to support a FANET.

The terrestrial infrastructure-aided control plane can provide a number of advantages, including mobility control of UAVs by gNBs so that topology control and routing can be done in a centralized manner. However, the coverage of gNBs may not be sufficient to communicate with all the UAVs when the density of gNBs is low. In addition, if there are a large number of UAVs, the bandwidth allocated to the control plane should increase, which is often undesirable due to the limited bandwidth. As a result, the terrestrial infrastructure-aided control plane should adapt to given conditions, e.g., the number of UAVs, coverage limit, bandwidth allocation, etc.

### 7.2. Mobility Management

Having a realistic mobility model for UAVs in FANETs is a crucial task. It represents the position, acceleration, and speed of UAVs in different time instances [24,25,160]. Although there are some mobility models defined for MANETs and VANETs which may be applicable to FANETs as well, the difference between FANETs and MANETs/VANETs should be further taken into account to have more FANET-oriented models. These differences include but are not limited to 3D moving area, power supply, safety, freedom in speed and directions, and so on.

### 7.3. Radio Resource Management

There are limited radio resources (e.g., bandwidth, power, storage, etc) for UAVs to operate efficiently when the number of users increases. It results in a need for radio resource

management and optimization to efficiently utilize available resources and provide services to the network's edge [161]. In addition, when UAVs are used to create flying hotspots, the radio resources should be carefully divided and allocated for communications between UAVs and ground users as well as communications between UAVs. This resource allocation to links between UAVs has to be considered with the routing of FANET itself and traffic from/to ground users in each UAV.

Furthermore, since the source of power in UAVs is mostly limited, i.e., UAVs are mainly powered by batteries, power efficiency through efficient resource management is especially important when UAVs are utilized in public safety applications [43]. Hence, having application-based resource optimization scenarios is a crucial research direction in future FANETs. This may be done according to different criteria to decrease the mission completion time, decrease total cost, increase scalability and survivability of the network.

### 7.4. Reconfigurable Intelligent Surfaces

Reconfigurable intelligent surface (RIS) is an emerging technology that can manipulate impinging electromagnetic waves to benefit wireless users [162,163]. One of the benefits of RIS is to scale up signal power and to reflect signals in the desired path by applying particular phase shifts [164]. A FANET equipped by RISs has the potential to be used for simultaneous wireless information and power transfer (SWIPT) [165]. Furthermore, in ultra-dense urban areas where many obstacles block the signal propagation in terrestrial networks, a FANET of RISs can enable the intelligent reflection from the sky to have a LoS transmission between terrestrial transceivers as shown in Figure 16. Hence, more research and development are required in the domain of RIS-assisted FANETs in the future.

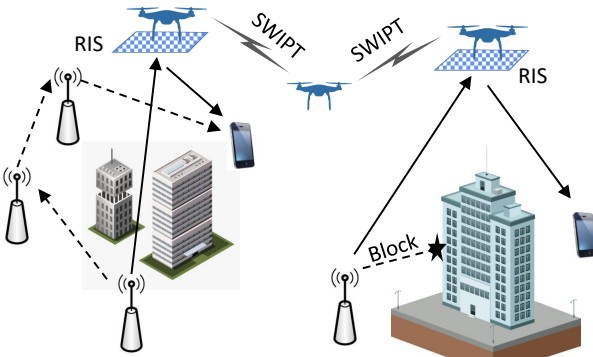

**Figure 16.** FANET can be equipped with RIS for reducing the number of backhaulings and/or overcoming the blockage in dense urban areas.

### 7.5. Advanced Antenna Technologies

The deployment of single omnidirectional antennas in UAVs is not an optimal case, especially in terms of power management. A directional antenna, on the other hand, may meet this problem with greater power efficiency by beamforming a signal solely towards a conic zone range rather than across all-around areas. It results in a significant reduction in power consumption up to $(2\pi\Phi - 1)$ times, where $\Psi \leq \pi$ is the half-power beamwidth [166]. However, since there is high mobility in a FANET topology, real-time and high-performing beam switching is a crucial task to make sure the data traffic is exchanged between mobile nodes effectively. Directional beamforming and beam switching are applicable by technologies such as MIMO [167–169] or steerable antennas [170]. Furthermore, the location information of the network should be accessible by the intended UAVs. This requires further research and studies to use narrow beam operations' beam sweeping characteristics to increase coverage and capacity while decreasing interference [58].

### 7.6. Feedback Based Retransmissions

Conventional feedback-based LTE retransmission techniques like automatic repeat request (ARQ) or hybrid ARQ (HARQ) might not be efficient in FANET since the channel conditions are mostly hostile and poor due to rapid variations in the network topology [171]. Indeed, these techniques are applicable when the channel variation is slow [172]. Likewise, the performance of traditional modulation schemes such as M-QAM begins to decay as the channel variation increases in FANET which affects the data transmission rates. Hence, having a type of adaptive modulation and coding (AMC) scheme for different channel conditions is profitable. Indeed, AMC can be considered a solution to the performance degradation of the system in poor channel conditions. Furthermore, compared to the conventional feedback-based approaches of ARQ/HARQ, AMC can help as an alternative to estimate the CSI and send feedback to the transmitter. This can also be a great help for power control on the transmitter side, since power management is an important issue in FANETs.

### 7.7. Time/Frequency/Space Dimensions

In FANETs, there are still serious challenges in the time, frequency, and space domains that should be addressed carefully. For instance, the time delay propagation of FANETs can vary from 1 ms to several milliseconds according to their altitudes. Therefore, transceivers should be able to dynamically adjust the uplink time advance parameters within a large range [3]. The time delay also impacts the HARQ-ACK process [173], although the HARQ-ACK can be disabled in specific scenarios at the cost of reliability. In the frequency domain, because of the fast and variable speed of a large number of UAVs, large Doppler frequency shifts may require complex frequency pre-compensation techniques at the receivers [83]. Moreover, there is a need for multiplexing and dynamic spectrum sharing to control the interference in highly varying network topologies. Likewise, in the space domain, frequent handover, location prediction, and trajectory optimization are challenging issues that require further research with respect to the specific application requirements.

### 7.8. Network Coding

Apart from feedback-based retransmission techniques in Section VII.F, the development of network coding strategies [117,174] for FANETs where conventional channel coding schemes, e.g., cyclic redundancy check (CRC), are constrained, can be beneficial. Network coding is an alternative to feedback-based reliable transmission techniques like HARQ. Indeed, a feedback-based approach can be problematic in FANETs since UAVs can move far from terrestrial and non-terrestrial nodes and re-transmissions in ARQ/HARQ impose large latency into the system due to long RTT [90]; thus, precluding agile development. Therefore, network coding can be applied over the data packets for packet-level coding and is expected to have a better performance in terms of throughput than conventional repetition-based coding. Hence, network coding is the potential solution that the system can leverage on the benefits of non-redundant packet re-submissions due to FANET challenging fading environments and long RTT. Indeed, a proposal of network coding strategies for packet-level coding can leverage existing UAVs for cost-effective FANETs and reliable connectivity while enhancing the system throughput, reducing delays, and constructing a more robust network.

## 8. Conclusions

Towards reaching the full potential of NTNs, in this article we have focused on the role of UAV networks, i.e., FANETs, catalyzing the integration between terrestrial and flying networks while provisioning various UAV-aided applications in 5G and beyond. In particular, we have identified that DTN-based routing methods and ML-assisted techniques are promising to further improve the connectivity and resource efficiency of FANETs, as partly evidenced by our LAROD-based FANET routing case study. Meanwhile, we have also discussed that there still remain a variety of research questions on mobility management,

privacy, advanced transmission techniques, and so forth. Based on this, investigating how to make DTN and ML-assisted FANETs compatible with other NTN components while jointly optimizing the integrated network resources could be an interesting avenue for future research.

**Author Contributions:** Conceptualization, M.N. and J.C.; methodology, M.N.; software, M.N.; validation, M.N., B.A.H. and S.W.L.; formal analysis, M.N.; investigation, M.N. and B.A.H. and S.K. and J.P. and S.W.L. and J.C.; resources, M.N. and B.A.H. and S.K. and J.P. and S.W.L. and J.C.; data curation, M.N.; writing—original draft preparation, M.N. and B.A.H. and S.K. and J.P. and S.W.L. and J.C.; writing—review and editing, M.N. and J.P. and S.W.L. and J.C.; visualization, M.N. and J.C.; supervision, J.C.; project administration, J.C.; funding acquisition, J.C. All authors have read and agreed to the published version of the manuscript.

**Funding:** This work was supported by the Institute of Information & communications Technology Planning & Evaluation (IITP) grant funded by the Korea government (MSIT) (No. 2021-0-00794, Development of 3D Spatial Mobile Communication Technology).

**Institutional Review Board Statement:** The study did not require ethical approval.

**Informed Consent Statement:** Not applicable.

**Data Availability Statement:** Not applicable.

**Conflicts of Interest:** The authors declare no conflict of interest.

## Abbreviations

The following abbreviations are used in this manuscript:

| | |
|---|---|
| 3GPP | 3rd Generation Partnership Project |
| A2A | Air-to-Air |
| A2G | Air-to-Ground |
| ACK | Acknowledgment |
| AI | Artificial Intelligence |
| ARQ | Automatic Repeat Request |
| CAGR | Compound Annual Growth Rate |
| CRC | Cyclic Redundancy Check |
| CSI | Channel State Information |
| DL | Deep Learning |
| DNN | Deep Neural Network |
| DQL | Deep Q-learning |
| DQN | Deep Q-Network |
| DTN | Delay Tolerant Network |
| eMBB | Enhanced Mobile Broadband |
| ESN | Echo State Networks |
| FANET | Flying Ad-hoc Network |
| FDD | Frequency Division Duplex |
| FL | Federated Learning |
| FLS | Flooding-based Location Service |
| G2A | Ground-to-Air |
| GEO | Geosynchronous/Geostationary Equatorial Orbit |
| gNB | Ground New Base Station |
| GPS | Global Positioning Systems |
| HAAP | High Altitude Aeronautical Platforms |
| HAP | High Altitude Platforms |
| HARQ | Hybrid Automatic Repeat Request |
| ICO | Intermediate Circular Orbit |
| IEEE | Institute of Electrical and Electronics Engineers |
| IMU | Inertial Measurement Unit |
| ISI | Inter-Symbol Interference |

| LEO | Low Earth Orbit |
| LTE | Long-Term Evolution |
| MANET | Mobile Ad-hoc Network |
| MEO | Medium Earth Orbit |
| ML | Machine Learning |
| mMTC | massive Machine Type Communications |
| mmWave | Millimeter-Wave |
| NOMA | Non-Orthogonal Multiple Access |
| NR | New Radio |
| RAN | Radio Access Network |
| RL | Reinforcement Learning |
| RTT | Round Trip Time |
| SCF | Store-Carry-and-Forward |
| SVM | Support Vector Machine |
| TDD | Time Division Duplex |
| TTL | Time To Live |
| UAS | Unmanned Aircraft Systems |
| URLLC | Ultra-Reliable and Low Latency Communications |
| UTM | UAS Traffic Management |
| VANET | Vehicular Ad-hoc Network |

## Appendix A. Topology-Based Routing Protocols

Below is the list of conventional topology-based routing protocols for FANETs [19,36,39] with respect to their proposed time.

1994 Destination Sequence Distance Vector (DSDV) [175]
1996 Dynamic Source Routing (DSR) [176]
1998 Zone Routing Protocol (ZRP) [177]
1998 Temporally Ordered Routing Algorithm (TORA) [178]
1999 Ad hoc on Demand Vector (AODV) [179]
1999 Hybrid Routing Protocol (HRP) [180]
2000 Fisheye-State Routing (FSR) [181]
2000 Multicast Ad hoc on Demand Vector (MAODV) [182]
2001 Optimised Link State Routing (OLSR) [183]
2002 Data-Centric Routing (DCR) [184]
2003 Sharp Hybrid Adaptive Routing Protocol (SHARP) [185]
2004 Topology Broadcast based on Reverse-Path Forwarding (TBRPF) [186]
2007 Load, Carry and Delivery (LCAD) [131]
2007 Time Slotted Ad hoc on Demand Vector (TS-AODV) (TS-AODV) [187]
2008 Better Approach to Mobile Ad Hoc Network (BATMAN)[188]
2008 Modified Optimised Link State Routing (MOLSR) [189]
2008 Hybrid Routing based on Clustering (HRC) [190]
2010 Directional Optimised Link State Routing (DOLSR) [191]
2011 Cartography-Enhanced Optimised Link State Routing (CE-OLSR) [192]
2011 Better Approach to Mobile Ad Hoc Network-Advanced (BATMAN-ADV) [193]
2011 BABEL [194]
2012 Contention Based Optimised Link State Routing (COLSR) [195]
2012 Ad hoc on Demand Vector Security (AODVSEC) [196]
2012 Mobility Prediction Clustering Algorithm (MPCA) [197]
2013 Predictive Optimised Link State Routing (POLSR) [198]
2013 Hybrid Wireless Mesh Protocol (HWMP) [199]
2013 Rapid-reestablish Temporally Ordered Routing Algorithm (RTORA) [200]
2014 Mobility and Load aware Optimised Link State Routing (ML-OLSR) [201]
2014 Multi-Level Hierarchical Routing (MLHR) [20]
2017 UAV-assisted routing (UVAR) [202].

## Appendix B. Position-Based Routing Protocols

Below is the list of conventional position-based routing protocols for FANETs [19,36,39] with respect to their proposed time.

2006 Ad Hoc Routing Protocol for Aeronautical Mobile Ad Hoc Networks (ARPAM) [203]
2008 Greedy-Random-Greedy (GRG) [204]
2009 Geographic Greedy Perimeter Stateless Routing (GPSR) [205]
2009 UAV Search Mission Protocol (USMP) [205]
2009 Greedy-Hull-Greedy (GHG) [206]
2010 Multipath Doppler Routing (MUDOR) [207]
2010 Greedy Distributed Spanning Tree Routing 3D (GDSTR-3D) [208]
2011 Reactive-Greedy-Reactive (RGR) [209]
2011 Greedy Geographic Forwarding (GGF) [210]
2011 Geographic Load-Share Routing (GLSR) [211]
2012 Geographic Position Mobility-Oriented Routing (GPMOR) [212]
2012 Mobility Prediction-Based Geographic Routing (MPGR)[213]
2014 Recovery Strategy for the Greedy Forwarding Failure (RSGFF) [214]
2014 Cross-Layer Link Quality and Geographical-Aware Beaconless [215]
2015 Connectivity-Based Traffic-Density Aware Routing Using UAVs for VANETs (CRUV) [118]
2016 UAV-Assisted VANET Routing Protocol (UVAR) [216]
2016 Position-Aware Secure and Efficient Routing Approach (PASER) [217]
2016 Secure UAV Ad Hoc Routing Protocol (SUAP) [218].

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
