# Peer review of "Non-Terrestrial Networks with UAVs: A Projection on Flying Ad-Hoc Networks"

_drones, doi:10.3390/drones6110334_

Round 1
Reviewer 1 Report
This paper discusses on NTN structure and key components while narrowing the topic down from a satellite perspective into UAV-based NTN, and presents a thorough literature study on the most recent advances in flying ad-hoc networks (FANET). It also investigates the FANET’s characteristic and advantages over existing approaches. However, the references should be doubly checked for their correctness, such as [63]’s and [64]’s URL. The “G.T.” in [63] and [64] also confuses the reviewer.
Author Response
Dear Editor and Reviewers,
We would like to take this opportunity to express our deep gratitude for your precious time and efforts spent on providing us with constructive comments and suggestions to improve our paper, entitled “Non-Terrestrial Networks with UAVs: A Projection on Flying Ad-Hoc Networks”. We carefully read all the comments and suggestions and addressed them in the revised manuscript.
Author response: We would like to thank the reviewer for this comment. We have now revised and rectified references [63] and [64].
Author action:
Page 27, Line 1032:
[63] 3GPP TR 22.822. Study on using satellite access in 5G. [Online]. Available: http:www.3gpp.orgftpSpecsarchive22_series 22.822/22822- 1040 g00.zip 2018. 1041
[64] 3GPP TR 38.811. Study on new radio to support non-terrestrial networks. [Online]. Available: http:www.3gpp.orgftpSpecs 1042 archive/38_series/38.811/38811-f00.zip 2018.
Acknowledgment:
Again, many thanks for your efforts invested in improving the paper.

Reviewer 2 Report
This review paper summarized the developments in non-terrestrial communication networks with UAVs. In general, this paper is well-written and can be considered for publication, if the following comments can be addressed appropriately:
1. In Section I, the concept of the NTN should be briefly mentioned.
2. In Section 4.2, the dynamically changing interference should be taken into account. Difference from traditional networks, the UAV networks can have clearer mobility models so that the time-varying interference can be predicted more accurately. For example, “Interference Prediction in Wireless Networks: Stochastic Geometry Meets Recursive Filtering” and “Interference Prediction in Mobile Ad Hoc Networks With a General Mobility Model.”
Author Response
Dear Editor and Reviewers,
We would like to take this opportunity to express our deep gratitude for your precious time and efforts spent on providing us with constructive comments and suggestions to improve our paper, entitled “Non-Terrestrial Networks with UAVs: A Projection on Flying Ad-Hoc Networks”. We carefully read all the comments and suggestions and addressed them in the revised manuscript.
Reviewer#2, Concern # 1: This review paper summarized the developments in non-terrestrial communication networks with UAVs. In general, this paper is well-written and can be considered for publication, if the following comments can be addressed appropriately:
- In Section I, the concept of the NTN should be briefly mentioned.
Author response: We would like to thank the reviewer for this comment. We agree with the reviewer. We have now revised the paper as follows:
Author action:
Page 1, Section 1, Line 1:
“Non-terrestrial networks (NTN) are to provide wireless connectivity from flying objects above ground usually from space and the stratosphere. In a nutshell, NTNs involve non-terrestrial flying objects such as satellites and unmanned aerial vehicles/drones (UAVs). Fig. 1 illustrates the concept of NTN.”
Reviewer#2, Concern # 2: In Section 4.2, the dynamically changing interference should be taken into account. Difference from traditional networks, the UAV networks can have clearer mobility models so that the time-varying interference can be predicted more accurately. For example, “Interference Prediction in Wireless Networks: Stochastic Geometry Meets Recursive Filtering” and “Interference Prediction in Mobile Ad Hoc Networks With a General Mobility Model.”
Author response: We would like to thank the reviewer for this comment. We agree with the reviewer. We have now revised the paper as follows:
Author action:
Page 12, Section 4.2, Line 415: “Furthermore, in contrast to conventional networks, UAV networks are capable of having more transparent mobility models, which enables more precise forecasting of time-varying interference. The authors in [ 89] proposed a technique to predict the level of interference that is applicable to such UAV networks. In addition, in [90 ], a general framework was proposed to forecast the interference behavior via analysis of the mean value and moment-generating function of the interference prediction.”
Ref. [89] Schmidt, J.F.; Schilcher, U.; Atiq, M.K.; Bettstetter, C. Interference prediction in wireless networks: Stochastic geometry meets recursive filtering. IEEE Transactions on Vehicular Technology 2021, 70, 2783–2793.
Ref. [90] Cong, Y.; Zhou, X.; Kennedy, R.A. Interference prediction in mobile ad hoc networks with a general mobility model. IEEE Transactions on Wireless Communications 2015, 14, 4277–4290.
Acknowledgment:
Again, many thanks for your efforts invested in improving the paper.

Reviewer 3 Report
In this paper, the authors investigate UAV networks integrated into terrestrial networks. Furthermore, they present the applications of FANETs and discuss the role of machine learning in FANETs.
Could you please add the simulation parameters for the case study?
It would be good if you could develop learning-based algorithms, e.g., reinforcement or deep learning for the simulation part.
Did you consider a minimum distance between UAVs for safety issues?
The following research works are missed in the paper focused on integrating UAVs into terrestrial networks for trajectory design and resource management problems using ML algorithms.
AH Arani, P Hu, Y Zhu, ‘’Re-envisioning space-air-ground integrated networks: Reinforcement learning for link optimization’’, ICC 2021
E. Kalantari, I. Bor-Yaliniz, A. Yongacoglu and H. Yanikomeroglu, "User association and bandwidth allocation for terrestrial and aerial base stations with backhaul considerations," IEEE (PIMRC), 2017.
Author Response
Dear Editor and Reviewers,
We would like to take this opportunity to express our deep gratitude for your precious time and efforts spent on providing us with constructive comments and suggestions to improve our paper, entitled “Non-Terrestrial Networks with UAVs: A Projection on Flying Ad-Hoc Networks”. We carefully read all the comments and suggestions and addressed them in the revised manuscript.
Reviewer#3, Concern # 1: In this paper, the authors investigate UAV networks integrated into terrestrial networks. Furthermore, they present the applications of FANETs and discuss the role of machine learning in FANETs.
Could you please add the simulation parameters for the case study?
Author response: We would like to thank the reviewer for this comment. We have now revised the paper and added the simulation parameters in “Table 3”.
Author action:
Pages 16 and 17, Section 4, Line 573:
“Simulation is carried out using the opportunistic network environment (THE ONE) simulator with simulation parameters given in Table 3. Random Waypoint mobility is considered. The rest of the parameters are the same as those in [24] unless otherwise specified.”
Reviewer#3, Concern # 2: It would be good if you could develop learning-based algorithms, e.g., reinforcement or deep learning for the simulation part.
Author response: We would like to thank the reviewer for this comment. We totally agree with the reviewer that the development of such learning-based algorithms would be beneficial. However, please note that such developments are out of the scope and space of this survey paper, and investigating this is differed to our future development works.
Reviewer#3, Concern # 3: Did you consider a minimum distance between UAVs for safety issues?
Author response: We would like to thank the reviewer for this comment. Yes. We have now clarified it as follows:
Author action:
Page 16, Section 4, Line 581: “The UAV nodes are supposed to be small, e.g., DJI mini2 20 × 20 cm2, and their minimum distance cannot be less than 1 m.”
Reviewer#3, Concern # 4: The following research works are missed in the paper focused on integrating UAVs into terrestrial networks for trajectory design and resource management problems using ML algorithms.
AH Arani, P Hu, Y Zhu, ‘’Re-envisioning space-air-ground integrated networks: Reinforcement learning for link optimization’’, ICC 2021
- Kalantari, I. Bor-Yaliniz, A. Yongacoglu and H. Yanikomeroglu, "User association and bandwidth allocation for terrestrial and aerial base stations with backhaul considerations," IEEE (PIMRC), 2017.
Author response: We would like to thank the reviewer for this comment. We have now revised the paper and included those research works as follows:
Author action:
Page 13, Section 4.3, Line 470: “In [ 106 ], a model was proposed to manage the radio resource among UAVs, and to optimize the trajectories of UAVs in the network by utilizing the tools of reinforcement learning.”
Ref. [106] Arani, A.H.; Hu, P.; Zhu, Y. Re-envisioning space-air-ground integrated networks: Reinforcement learning for link optimization. In Proceedings of the ICC 2021-IEEE International Conference on Communications. IEEE, 2021, pp. 1–7.
Page 19, Section 6.2, Line 669: “In [155 ], a ML algorithm was proposed focusing on the integration of UAVs into terrestrial networks to find efficient resource allocation along with the optimized 3D locations of UAVs to maximize the sum logarithmic rate of the users.”
Ref. [155] Kalantari, E.; Bor-Yaliniz, I.; Yongacoglu, A.; Yanikomeroglu, H. User association and bandwidth allocation for terrestrial and aerial base stations with backhaul considerations. In Proceedings of the 2017 IEEE 28th Annual International Symposium on Personal, Indoor, and Mobile Radio Communications (PIMRC). IEEE, 2017, pp. 1–6.
Acknowledgment:
Again, many thanks for your efforts invested in improving the paper.

Reviewer 4 Report
This paper presented an overviews the key components of NTN while highlighting the significance of emerging UAV networks. In addition, both existing and emerging applications of the FANET are explored. Next, it provides key recent findings and the state-of-the-art of FANETs while examining various routing protocols based on cross-layer modeling. The work is well organized and appropriately carried out. This work definitely worth publication. Should be supported the study by more new references (2021 and 2022). How could/should future studies improve the model?
Author Response
Dear Editor and Reviewers,
We would like to take this opportunity to express our deep gratitude for your precious time and efforts spent on providing us with constructive comments and suggestions to improve our paper, entitled “Non-Terrestrial Networks with UAVs: A Projection on Flying Ad-Hoc Networks”. We carefully read all the comments and suggestions and addressed them in the revised manuscript.
Author response: We would like to thank the reviewer for this comment. We have now revised the paper and reviewed more recent references. Nevertheless, we have tried our best to review the flagship references and may miss some of the most recently published articles, and investigating them is differed to our future research works.
Author action:
Ref. [3] Liu, J.; Peng, S.; Jiang, Z.; She, X.; Chen, P. Operation and Key Technologies in Space-Air-Ground Integrated Network. In Proceedings of the 2022 International Wireless Communications and Mobile Computing (IWCMC), 2022, pp. 1311–1316. https://doi.org/10.1109/IWCMC55113.2022.9825014.
Ref. [5] Priyadarshini, I.; Bhola, B.; Kumar, R.; So-In, C. A Novel Cloud Architecture for Internet of Space Things (IoST). IEEE Access 2022, 10, 15118–15134.https://doi.org/10.1109/ACCESS.2022.3144137.
Ref. [104]. Yanmaz, E. Positioning aerial relays to maintain connectivity during drone team missions. Ad Hoc Networks 2022, 128, 102800.
Ref. [109] You, W.; Dong, C.; Wu, Q.; Qu, Y.; Wu, Y.; He, R. Joint task scheduling, resource allocation, and UAV trajectory under clustering for FANETs. China Communications 2022, 19, 104–118.
Ref. [110] Singh, R.; Qu, C.; Esquivel Morel, A.; Calyam, P. Location Prediction and Trajectory Optimization in Multi-UAV Application Missions. In Intelligent Unmanned Air Vehicles Communications for Public Safety Networks; Springer, 2022; pp. 105–131.
Ref. [111] Albu-Salih, A.T.; Khudhair, H.A.; Hilal, O.M. Data acquisition time minimization in FANET-based IoT networks. Kuwait Journal of Science 2022, 49.
Ref. [112] Da Silva, I.D.; Caillouet, C.; Coudert, D. Optimizing FANET deployment for mobile sensor tracking in disaster management scenario. In Proceedings of the 2021 International Conference on Information and Communication Technologies for Disaster Management (ICT-DM). IEEE, 2021, pp. 134–141.
Ref. [166] Shah, Z.; Naeem, M.; Javed, U.; Ejaz, W.; Altaf, M. A compendium of radio resource management in UAV-assisted next generation computing paradigms. Ad Hoc Networks 2022, 131, 102844.
Ref. [167] Nemati, M.; Maham, B.; Pokhrel, S.R.; Choi, J. Modeling RIS empowered outdoor-to-indoor communication in mmWave cellular networks. IEEE Transactions on Communications 2021, 69, 7837–7850.
Ref. [173] Koul, S.K.; Wani, Z. Beam Switching and Wide-Scan Antenna Array. In Novel Millimetre Wave Antennas for MIMO and 5G Applications; Springer, 2021; pp. 73–92.
Ref. [174] Chen, K.; Liu, D.; He, X. Fast Beam Switching Based on Machine Learning for MmWave Massive MIMO Systems. In Proceedings of the International Conference on Artificial Intelligence for Communications and Networks. Springer, 2021, pp. 19–29.
Ref. [177] Vaezi, M.; Azari, A.; Khosravirad, S.R.; Shirvanimoghaddam, M.; Azari, M.M.; Chasaki, D.; Popovski, P. Cellular, wide-area, and non-terrestrial IoT: a survey on 5G advances and the road toward 6G. IEEE Communications Surveys & Tutorials 2022, 24, 1117–1174.
Ref. [178] Qi, Y.; Deng, H.; Liu, M.; Li, Z. On the Standardization of Key Enabling Technologies in Non-Terrestrial Network Air-interface Design. In Proceedings of the 2021 IEEE/CIC International Conference on Communications in China (ICCC Workshops). IEEE, 2021, pp. 343–348.
Ref. [179] Choi, J.; Ding, J. Network coding for k-repetition in grant-free random access. IEEE Wireless Communications Letters 2021, 10, 2557–2561.
Acknowledgment:
Again, many thanks for your efforts invested in improving the paper.
